# Pediatric Lichen Myxedematosus: A Diagnostic and Management Challenge

**DOI:** 10.3390/children9070949

**Published:** 2022-06-25

**Authors:** Kelly K. Barry, Diana B. Reusch, Birgitta A. R. Schmidt, Elena B. Hawryluk

**Affiliations:** 1Tufts University School of Medicine, Boston, MA 02111, USA; kelly.barry@childrens.harvard.edu; 2Dermatology Section, Division of Immunology, Boston Children’s Hospital, Harvard Medical School, Boston, MA 02115, USA; diana.reusch@childrens.harvard.edu; 3Department of Pathology, Boston Children’s Hospital, Harvard Medical School, Boston, MA 02115, USA; birgitta.schmidt@childrens.harvard.edu; 4Department of Dermatology, Massachusetts General Hospital, Harvard Medical School, Boston, MA 02114, USA

**Keywords:** lichen myxedematosus, papular mucinosis, nodular lichen myxedematosus, pediatric, juvenile

## Abstract

Localized lichen myxedematosus (LM) is a rare, idiopathic mucinosis characterized by dermal mucin deposition and variable fibroblast proliferation. Nodular lichen myxedematosus, a clinicopathologic subtype of localized LM, is exceedingly rare in pediatric patients with only three prior cases reported. Understanding of LM in pediatric patients is limited by the rarity of the disease, and diagnosis is complicated by overlapping clinical and histopathologic features. There is no standardized treatment for localized LM and treatment is largely dictated by a patient’s desire to minimize cosmetic disfigurement. This case series reports two additional patients with juvenile nodular lichen myxedematosus, highlights the limitations of existing diagnostic criteria, and describes successful treatment of one patient with intralesional triamcinolone.

## 1. Introduction

Localized lichen myxedematosus (LM) is a rare, idiopathic, mucinosis characterized by dermal mucin deposition and variable fibroblast proliferation. The diagnosis of LM and determination of subtype can be challenging due to overlapping clinical and histopathologic features. In the classification system proposed by Rongioletti and Rebora in 2001, localized LM is distinguished from systemic scleromyxedema by the absence of systemic symptoms, monoclonal gammopathy, and sclerodermoid features [1]. Thyroid disease is considered exclusionary for both LM and scleromyxedema. A third category of atypical LM is recognized for cases that do not meet criteria for scleromyxedema or localized LM. A revised classification system was later proposed by Nofal et al. (2017) in which patients must present with waxy, firm papules demonstrating diffuse dermal mucin deposition and fibroblast proliferation, but generalized cutaneous eruption, systemic symptoms, and thyroid disease are not exclusionary for diagnosis (Table 1) [2]. In the new classification system, Nofal et al. (2017) describe a “pure cutaneous subtype” which encompasses previously defined localized LM subtypes including discrete, infantile, acral persistent, self-healing, and nodular LM. Among the subtypes of localized LM, nodular lichen myxedematosus (NLM) is exceedingly rare in the pediatric population, with only 3 prior cases identified in the literature [3,4,5]. Herein we report two pediatric patients with a clinicopathologic diagnosis most consistent with NLM, adding to the limited existing literature of this rare disease and highlighting the limitations of current diagnostic criteria.

## 2. Case Descriptions

### 2.1. Case 1

A 13-year-old boy presented with asymptomatic lesions on his abdomen, arms, and legs that erupted over five months. The patient reported chronic intermittent abdominal pain and diarrhea, but review of systems was otherwise negative, and he denied any joint pains. History included infantile eczema and irritable bowel syndrome, and family history was negative for similar papules. Laboratory workup revealed alkaline phosphatase 287 U/L, white blood cells 12,300 cells/uL, and hemoglobin 12.6 g/dL. Paraprotein screening and celiac disease antibody testing were normal. He also underwent consultation by a pediatric gastroenterologist, with unremarkable workup.

Examination revealed approximately 30 pink-yellow nodulopapular lesions (8–15 mm) clustered on the patient’s arms and legs (Figure 1a), and distributed more sparsely on his torso. The most distal leg lesions were flatter and more pink-brown than the others, despite no previous intervention. Forearm shave biopsy revealed dermal mucin deposition separating the collagen bundles with an increased number of fibroblasts and a histiocytic infiltrate (Figure 1b). PAS stain was negative for fungal forms. Histopathologically, this patient exhibits features of both NLM and self-healing juvenile cutaneous mucinosis (SHJCM), however clinical presentation favors a diagnosis of NLM.

After unsuccessful treatment with topical tacrolimus, three cosmetically bothersome lesions were injected with intralesional triamcinolone (40 mg/mL). Treated lesions demonstrated marked flattening within one month (Figure 2), with no change observed in non-treated lesions. Several lesions persist, and others have resolved, at 7 years after initial photos.

### 2.2. Case 2

A 7-year-old girl presented with intermittently pruritic papules and nodules on her bilateral legs, lower back, and dorsal left hand that erupted over 16 months. Her review of systems was negative, and her history was unremarkable. Family history was notable for hypothyroidism in her mother, a large congenital nevus and café au lait macule in her father, and cancer in both grandparents (paraganglioma with positive SDHA mutation in paternal grandmother and chronic lymphocytic leukemia in her paternal grandfather) and was negative for similar cutaneous papules. Laboratory workup was notable for antinuclear antibody 1:160, speckled, LDH 329 U/L, and thyroid stimulating hormone 6.67 mcunit/mL. Paraprotein screening and celiac disease screening were normal. She was evaluated by a pediatric endocrinologist and repeat thyroid testing returned normal.

Examination revealed skin-colored papules and nodules (2–10 mm) clustered around the left anterior knee (Figure 3) in addition to sparsely distributed skin-colored papules and nodules on the right lower leg, left lower back, and left dorsal hand. Medial thigh shave biopsy revealed superficial and mid-dermal spindle cell proliferation with increased dermal mucin deposition (Figure 4A,B). An Alcian blue stain demonstrated a prominent increase in dermal mucin (Figure 4C). Histopathologically, this patient exhibits features of both NLM and SHJCM, however clinical presentation favors a diagnosis of NLM.

Treatment of her lesions with fluocinonide and mometasone creams was unsuccessful. Given that her lesions are asymptomatic, further treatment was deferred. New lesions continue to develop without spontaneous resolution over 2-year follow-up.

## 3. Discussion

Nodular LM (NLM) is a subtype of localized LM with only three previously reported pediatric cases [3,4,5]. The average age of previously published pediatric NLM is 10 years (range 6–18 years) and two of the three reported patients were male. These patients had asymptomatic papules and nodules of the upper arms with variable involvement of the trunk and lower limbs. No patients reported prodromal or associated systemic symptoms, laboratory studies were all unremarkable with absent monoclonal gammopathy, and the diagnosis was confirmed by biopsy [3,4].

Our patients developed lesions at ages 12 and 5, with a similar clinical presentation, though patient 2 had more prominent lesions on her lower extremities, rather than upper. The diagnosis of NLM was rendered based on the presence of histopathologic features including dermal mucin deposition with variable fibroblast proliferation in addition to clinical features including (1) an absence of systemic symptoms or lab abnormalities suggestive of scleromyxedema or thyroid-related myxedema, (2) lesion morphology and (3) lack of spontaneous clinical resolution. Other subtypes of localized LM, such as cutaneous mucinosis of infancy, acral persistent papular mucinosis, and discrete papular mucinosis do not fit clinically [6]. Mucinous nevus was considered; however, mucinous nevi typically present unilaterally in a zosteriform distribution and exhibit characteristic band-like deposition of mucin in the superficial dermis [7,8]. Notably, SHJCM which can also present with generalized papules and nodules, remains on the differential, however, lesions persist without evidence of resolution to date.

Diagnostic criteria for LM have undergone several iterations and were recently revised in 2017 (Table 1) [2]. Prodromal symptoms of SHJCM, which include periorbital edema, asthenia, arthralgia, and night sweats can be difficult to distinguish from the extracutaneous manifestations of scleromyxedema [9]. However, the systemic manifestations of scleromyxedema tend to be more severe and frequently include neurologic (e.g., carpal tunnel syndrome, sensory and motor neuropathy), cardiovascular (e.g., congestive heart failure, ischemia, heart block), and gastrointestinal (dysphagia) manifestations [1]. Nofal et al. (2017) propose a clinical severity grading system to spare extensive workup and treatment for patients with limited pure cutaneous involvement who are more likely to experience a benign clinical course [2]. While this is practically useful, cosmetic disfigurement universally remains an important issue; therefore, identifying self-healing presentations is key to counseling regarding prognosis and risks of management options.

SHJCM and NLM are both recognized as having a benign clinical course, though Luchsinger et al. (2018) reported two patients with a presumed diagnosis of SHJCM who had incomplete resolution with progression to fibroblastic and autoinflammatory rheumatism [9]. Prodromal symptoms, head or scalp involvement, and proliferative fasciitis-like changes in nodules favor the SHJCM subtype. However, distinction is primarily determined by lesion resolution, which is impractical for clinical diagnosis and patient counseling. Spontaneous resolution of SHJCM typically occurs within 2–8 months [9] and in rare circumstances several years [10,11,12,13]. Notably, treatment response may confound conclusions about the natural history, and previously reported cases of NLM may not have included enough follow-up to observe improvement.

Distinguishing SHJCM from NLM based on histopathologic features is rarely feasible. Dermal mucin deposition has failed to show a consistent pattern in either subtype [12,14,15]. Nodular lesions of SHJCM may exhibit proliferative fasciitis-like features or lobular panniculitis with increased mucin deposition [9], so biopsy of a deep or large lesion is advantageous. However, perivascular and periadnexal inflammatory infiltrate can also be observed in NLM [3,4,5].

There is no standardized treatment for localized LM. Treatment of localized LM is complicated by the potential for spontaneous resolution, lack of efficacy, and risks of systemic treatments; the decision to treat is largely dictated by a patient’s desire to minimize cosmetic disfigurement. Previously described treatment modalities include topical and systemic corticosteroids, psoralen + ultraviolet A, and topical retinoids [2,4,14]. Patient 1 of our report demonstrated improvement with intralesional triamcinolone, which in addition to hyaluronidase injection has been successfully reported for treating NLM [4,14].

In summary, the diagnosis of NLM is favored for both of our patients, though future spontaneous resolution would instead support a diagnosis of SHJCM. In the pediatric literature, there is a paucity of case reports, long-term follow up data, guidelines for systemic workup, and management options for LM. Notably, existing diagnostic criteria were developed primarily based on adult disease presentation and may not be generalizable to pediatric patients.

## Figures and Tables

**Figure 1 children-09-00949-f001:**
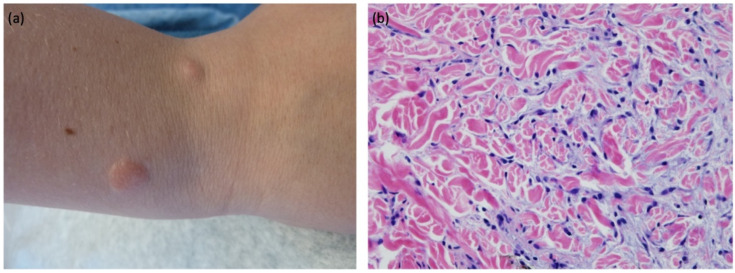
Right distal forearm erythematous, firm nodules (**a**) with H&E, original magnification × 40 (**b**) showing dermal mucin deposition separating the collagen bundles with increased number of fibroblasts as well as histiocytes.

**Figure 2 children-09-00949-f002:**
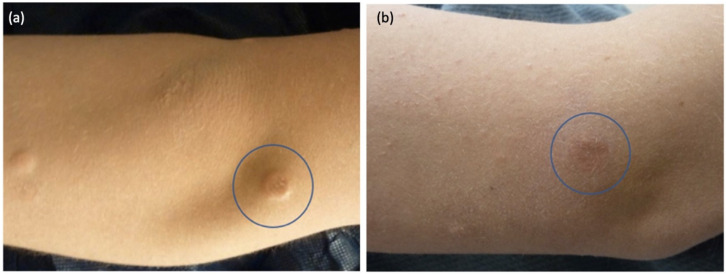
Right arm pink-yellow papule, before (**a**) and after (**b**) intralesional triamcinolone.

**Figure 3 children-09-00949-f003:**
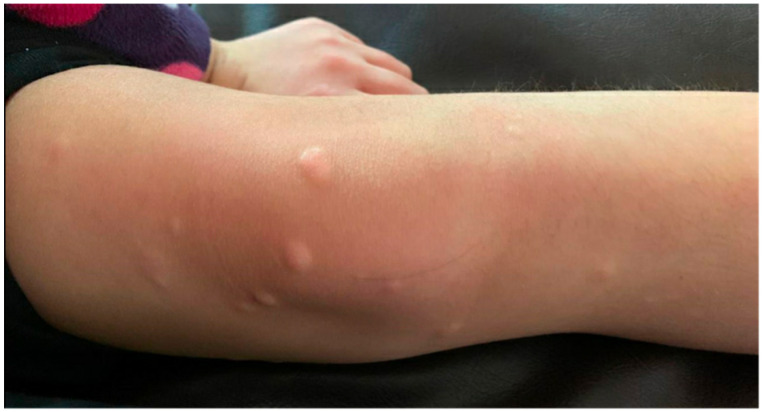
Right anterior knee skin-colored, firm papules and nodules.

**Figure 4 children-09-00949-f004:**
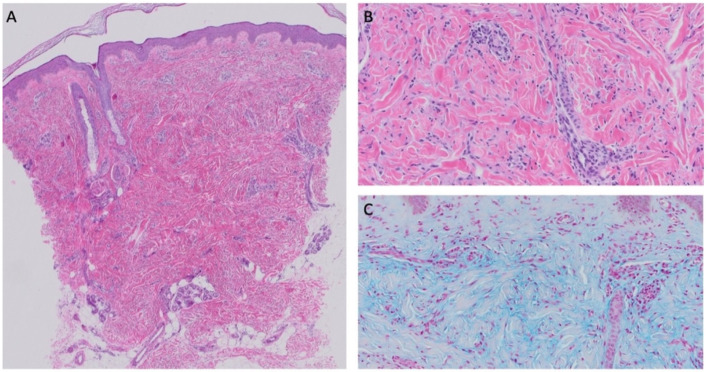
H&E, original magnification × 4 (**A**) and H&E original magnification × 20 (**B**) showing superficial and mid-dermal spindle cell proliferation with increased dermal mucin deposition. H&E, original magnification × 20 with Alcian blue stain highlights the dermal mucin deposition (**C**).

**Table 1 children-09-00949-t001:** A comparison of the 2001 lichen myxedematosus diagnostic criteria proposed by Rongioletti and Rebora and the updated 2017 diagnostic criteria proposed by Nofal et al.

Rongioletti and Rebora (2001)	Nofal et al. (2017)
Scleromyxedema Generalized papular and sclerodermoid eruption Microscopic triad (mucin deposition, fibroblast proliferation, fibrosis) Monoclonal gammopathy Absence of thyroid disorder	Constant Features (Scleromyxedema or Pure Cutaneous) Firm, waxy, closely set papules that may coalesce into indurated nodules or plaques Diffuse dermal mucin deposition and fibroblast proliferation with or without fibrosis
Localized LM Papular eruption (or nodules and/or plaques due to confluence of papules) Mucin deposition with variable fibroblast proliferation Absence of monoclonal gammopathy Absence of thyroid disorder	Variable Associated Features (Scleromyxedema or Pure Cutaneous) Monoclonal gammopathy Thyroid disorder HIV and hepatitis C virus infections Peripheral eosinophilia Thymic carcinoma and hepatocellular carcinoma
Clinicopathologic Subtypes: self-healing (juvenile and adult variant) *, discrete papular, acral persistent, papular mucinosis of infancy, nodular	
* Clinical criteria for juvenile variant include: (1) young age at presentation (2) acute eruption of papules on the face, neck, scalp, abdomen, thighs (3) variable development of nodules on face and periarticular regions (4) variable association with fevers, arthralgias, weakness (5) spontaneous resolution, typically within weeks to months	
Atypical Subtypes of LM Scleromyxedema without monoclonal gammopathy Localized LM with monoclonal gammopathy and/or systemic symptoms other than HIV infection Localized LM with mixed features of different subtypes Other nonspecified cases	Distinguishing Features Systemic (Scleromyxedema): Disabling, sometimes fatal localized or generalized cutaneous lesions Systemic manifestations
	Pure Cutaneous Subtype: Localized or limited skin involvement such as discrete, acral persistent, self-healing, and nodular forms or any emerging localized presentation Generalized or extensive skin involvement, without systemic manifestations

LM = lichen myxedematosus.

## Data Availability

Not applicable.

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
