# Peer review of "Pediatric Lichen Myxedematosus: A Diagnostic and Management Challenge"

_children, 2022, doi:10.3390/children9070949_

Round 1

Reviewer 1 Report

An interesting case report about a very rare dermatological condition. I think the paper is eligible to be published in its current form.

Reviewer 2 Report

The case report presented by authors is very informative. Considering the paucity of available case reports, authors raised a valid concern about the diagnostic criteria of NLM and SHJCM and it is arguable in adult and pediatric populations. Following comments are suggested for the manuscript.

1)      Line no. 61: There is irregularity in the text. Text mentioning “increased number of fibroblasts as well as histiocytes highlighted by CD163 immunohistochemistry staining (Figure 1b).” but figure 1b is H & E stain. No data for CD163 staining, please incorporate same or rephrase the sentence. Have authors performed Alcian blue staining in any of the cases, particularly in case 2?

2)      Abbreviation should be used cautiously in the manuscript, should be expanded wherever first used e.g. SHJCM (self-healing juvenile cutaneous mucinosis). Diagnostic (differential) criteria of SHJCM should also be mentioned in the Table 1 (if possible).

3)      Line number 67 “Lesions persist five years later.” Have authors followed up patient for 5 years. If so, Photograph of same to be incorporated for better interpretation.

4)      Line no. 88 and 89, No data to support CD34 and CD117, should be given as supplementary information if permissible by journal. 
